# Measuring Productivity, Perceived Stress and Work Engagement of a Nationally Delivered Workplace Step Count Challenge

**DOI:** 10.3390/ijerph19031843

**Published:** 2022-02-06

**Authors:** Gemma C. Ryde, Simone A. Tomaz, Krissi Sandison, Carl Greenwood, Paul Kelly

**Affiliations:** 1Institute of Cardiovascular and Medical Sciences, College of Medical Veterinary and Life Sciences, University of Glasgow, 126 University Place, Glasgow G12 8TA, UK; simone.tomaz@stir.ac.uk; 2Division of Sport, Faculty of Health Sciences and Sport, University of Stirling, Stirling FK9 4LA, UK; krissi.sandison@nhs.scot; 3NHS Shetland, Upper Floor Montfield, Burgh Road, Lerwick ZE1 0LA, UK; 4Paths for All, Kintail House, Forthside Way, Stirling FK8 1QZ, UK; carl.greenwood@pathsforall.org.uk; 5Physical Activity for Health Research Centre (PAHRC), The University of Edinburgh, Edinburgh EH8 8AQ, UK; p.kelly@ed.ac.uk

**Keywords:** physical activity, steps, walking, workplace, stress, productivity, work engagement

## Abstract

Workplace step count challenges show promise with regard to increasing physical activity, with walking linked to many positive physical and mental health benefits. Few studies have investigated their effects on work-related outcomes. The aim of this study was to describe (1) the process of collecting work-related outcomes as part of a real-world workplace intervention, the Step Count Challenge, and (2) report step counts and work-related outcomes (productivity, perceived stress and work engagement) during the Step Count Challenge. This pre-post study was conducted as part of a four-week 2018 National Step Count Challenge (SCC) in Scotland, UK. A survey was administered to collect self-reported steps, productivity (HWQ), perceived stress (Cohen Perceived Stress Scale) and work engagement (UWES) on the week prior to the intervention starting (baseline), week 1 and week 4. Process data such as recruitment and response rates were monitored throughout. Of 2042 employees who signed up to the SCC, baseline data were reported for 246 (12% of total; mean age 42.5 years, 83% female). Process data suggest low uptake to the study and poor compliance between surveys time points. Preliminary data suggest positive changes in step counts (3374 steps/day by week 4), stress and productivity. No changes in work engagement were found. The findings highlight the need to integrate research more effectively into real-world interventions, including a true baseline period. The Step Count Challenge may have positive change on some work-related outcomes warranting further investigation into how robust research designs can be delivered without negatively influencing real-world interventions.

## 1. Introduction

Over the past 50 years physical activity has decreased dramatically, coupled with an increase in sedentary behavior [1,2]. Time use data suggest this change in physical activity is largely attributable to a decrease in occupational physical activity [1,2]. With advancements in technology, many workplaces have shifted towards sedentary environments with predominantly desk-based occupations. Research suggests 42% of men and 47% of women report their jobs as mostly sitting and that fulltime office employees can sit for approximately six hours per day at work [3,4]. Very little movement occurs during work hours in these jobs with some data suggesting as little as 4 to 17 min of moderate to vigorous activity accumulated whilst at work [3,5]. With 76% of UK adults aged 16–64 in employment, too much sitting and not moving at work effects a large percentage of the UK population [6].

This low level of movement at work is a concern given that physical activity is strongly associated with improved physical, social, and mental wellbeing [7]. For workplaces and organisations, having employees who are physically active can also be beneficial for work-related outcomes as well as for the individuals themselves. For example, employees who are regularly active have been shown to be more productive [8,9,10,11,12] and employers who introduce workplace physical activity initiatives report positive benefits such as improved job satisfaction and mood [8,12,13]. Conversely, low levels of physical activity in employees are associated with many poor work-related outcomes including reduced productivity (quality and quantity of work) and increased absenteeism, health care costs, disability, work impairment, and musculoskeletal problems [14,15,16]. Addressing employees’ physical activity is therefore of significance not only for individual employees, but for workplaces as well.

Physical activity interventions delivered through the workplace could in part help to address the issue of inactivity at work. Whilst reviews of such interventions have shown mixed findings on their effects on physical activity when assessed collectively, there is evidence to suggest that some interventions might be more valuable than others in the workplace setting. One review reported an overall small effect size when all workplace physical activity interventions were assessed together but suggested that worksite walking interventions were nearly four times as beneficial than other forms of physical activity intervention in the workplace setting [17]. This is encouraging as walking as a specific form of physical activity has been shown to have significant benefits to both physical [18] and mental health [19]. Workplace walking interventions therefore show promise for increasing physical activity at work with potential to improve employee health outcomes.

However, few studies on workplace-walking interventions have reported the effectiveness to improve work-related outcomes. In a review addressing the effect of workplace health-enhancing physical activity initiatives on productivity [20], only eight studies were included, one of which addressed walking [21]. This one study on walking showed that a workplace walking intervention could positively influence work productivity in employees with low initial steps counts [21]. In a more recent review of workplace pedometer intervention by Freak-poli et al. 2020, 14 studies were included with none measuring work-related outcomes [22]. When looking specifically at stress as an outcome of interest to workplaces, a review by Chu et al. 2014. on workplace physical activity interventions for mental health benefits reported no studies that has addressed workplace walking and its effects on stress [23]. This overall lack of research into workplace walking interventions and work-related outcomes is surprising given that qualitative research suggests showing either improvements or no loss of productivity would be beneficial for employers when making the decision whether or not to invest in physical activity interventions during worktime [24]. Therefore, research on workplace walking interventions and work-related outcomes is needed to help evidence why investment (employees time and potentially organizational financial investment) in these types of interventions by employers is warranted.

The Step Count Challenge (https://www.stepcount.org.uk/) is a bi-annual, nationally delivered online workplace physical activity challenge delivered by Scottish Charity Paths for All and is a key aspect of Scotland’s National Walking Strategy. The Step Count Challenge has been running since 2013, with the spring and autumn challenges attracting on average 5000 participants each year. This shows that it is a sustainable and wide-reaching intervention. Whilst it is arguably more challenging to conduct research to determine effectiveness of real-world interventions (as opposed to more controlled research environments), this sort of study is essential in order to ensure research has wider societal impact [25]. The aim of this study was to describe (1) the process of collecting work-related outcomes as part of a real-world workplace step count challenge, and (2) report physical activity levels (step counts) and work-related outcomes (productivity, perceived stress, and work engagement) during the Step Count Challenge.

## 2. Materials and Methods

This study used a pre-post, observational design. Data were collected in October 2018. Ethics was approved by the University of Stirling General University Ethics Panel Ref: GUEP519.

### 2.1. Participants and Recruitment

Participants were employees from organizations throughout Scotland who had chosen to take part in the 2018 autumn workplace Step Count Challenge. The recruitment to the Step Count Challenge was coordinated by Paths for All and included emails to previous participants of the Step Count Challenge, existing general contacts, social media advertising through Facebook (including paid adverts), Twitter, Instagram and blog posts; and taking promotional materials to events and conferences. For the present study, 2042 employees signed up to the Step Count Challenge (1206 new users and 836 previous challenge users). Those who signed up to the challenge then received an introductory welcome blog with a section dedicated to the current study including a link to the survey with the participant information sheet and consent form.

### 2.2. Intervention

The Step Count Challenge is a bi-annual challenge run in the spring (May—eight weeks) and autumn (October—four weeks) each year in Scotland (https://www.stepcount.org.uk). The current study reports on the 2018 autumn challenge. The challenge is delivered through an online platform where teams of five employees with a nominated captain sign up at a cost of GBP 30 (USD ±40) per team. Each team member has their own profile page and dashboard where they can enter steps, monitor weekly step goals, and communicate with their team. They can also see their team members progress and how the overall teams’ steps compare with the wider challenge communities. For those who choose to take part in non-walking activities (e.g., cycling, swimming, running were other options for the 2018 challenges) the website gives you the opportunity to convert these into steps and add it towards your total. For example, for cycling a 1:4 ratio is used with four miles of cycling equating to one mile of walking. For this challenge 93% of all activities recorded were walking, 4% cycling, 2% running and 1% swimming. After the first week of the challenge where no step goal was set, individual step goals increased by an additional 1500 steps per day based on their week 1 step counts for the following 3 weeks with the aim to reach this goal on 3 days in week 2 and then 5 days in weeks 3 and 4 (Figure 1). Additional motivation was provided through daily emails with information on the individuals step counts and goal progress, weekly blogs and competitions including photo competitions, and step count champion awards with merchandise and vouchers presented to winners.

### 2.3. Survey

The survey was delivered at three time points through Jisc Online Surveys https://www.onlinesurveys.ac.uk/ (Figure 1). Each time point aimed to capture data for the previous 7 days in line with the time periods specified in the majority of validated questionnaires used. The first was delivered on the Monday of week one of the challenge, (reporter hereafter as ‘baseline’), the second on the Monday of week two (reporter hereafter as week 1) and the third on the Monday after the final week challenge (reporter hereafter as week 4). The survey comprised of four sections: Demographic information, steps, productivity, perceived stress and work engagement, which are detailed further below.

#### 2.3.1. Demographic Information

Questions relating to participant demographics included year of birth, gender, hours at work that week, number of hours your employer expected you to be at work that week, annual salary, employment status and sector. These data were only collected as part of the baseline survey.

#### 2.3.2. Steps

Total steps for the previous week and the number of days steps were recorded was reported. Whilst steps are entered online by employees as part of the Step Count Challenge, at the time of this study there was no method for automatically transferring these data for research purposes. No specific device is recommended for collecting step counts and employees can use their own activity trackers or mobile phone, or purchase step counters from Paths for All. In house, unpublished evaluation conducted by Paths for All for the spring 2018 challenge suggested that 70% employees used an activity tracker, 21% use a smartphone app and 9% used a pedometer. Employees that completed the survey were put into a prize draw after each survey with the opportunity to win a FitBit for each survey they completed (three chances to win in total).

#### 2.3.3. Productivity

The Health and Work Questionnaire (HWQ) measures employee productivity and health [26]. The survey consists of 24 questions (3 with sub-questions) used to create an overall HWQ score and six subscales; productivity, concentration/focus, supervisor relations, impatience/irritability, work satisfaction, and non-work satisfaction. Questions are rated on a 10-point Likert scale from a negative rating (1) to a positive rating (10). Scores for each subscale are calculated from the sum of responses to all the items comprising the subscale divided by the number of items in the subscale. An overall HWQ score is produced from the average of all six subscales. This overall HWQ score will be referred to as ‘total productivity’ as opposed to the subscale item which will remain as ‘productivity’. Scores closer to 10 indicate desirable productivity outcomes.

#### 2.3.4. Perceived Stress

Perceived stress was measured using the 10 item Cohen-Perceived Stress Scale [27]. This instrument is designed to measure psychological stress and to evaluate the stressful situations of daily life by measuring the degree an individual feels their life has been unpredictable, uncontrollable and overloaded during the past month [28]. Questions were rated on a five-point Likert scale ranging from never (0) to almost always (4) with the ratings for each question summed to create an overall stress score. Higher scores indicate more perceived stress with the possible range of summed scores ranging from zero to 40. Scores of 13 are classified as average stress levels and scores of 20 or above indicate high stress levels [29].

#### 2.3.5. Work Engagement

The Utrecht Work Engagement Scale (UWES) measures work engagement, which can be described as the opposite of burnout [30]. The UWES-17 consists of 17 statements used to create an overall engagement score and three subscales: vigor, absorption, and dedication. Statements are rated on a 7-point Likert scale from never had this feeling (0) to always every day (6). Scores for each subscale are calculated from the sum of responses to all the items within that subscale, divided by the number of items in the subscale. An overall engagement score is produced from the average of all 17 subscales. Normative scores range from very low to very high, with very low scores for vigor, dedication, absorption and total engagement reported as ≤2.17, ≤1.60, ≤1.60 and ≤1.93, respectively, and very high scores reported as ≥5.61, ≥5.80, ≥5.36 and ≥5.54, respectively.

### 2.4. Analysis

All survey data were downloaded into Excel (2016), scored according to survey user manuals where applicable and coded. Data were analysed using STATA (v.17, STATA Corp, College Station, TX, USA). Basic descriptive statistic and plots were used to visualize the data and check for outliers. For step data, any outliers were crosschecked against all three-time points to assess if data were entered in error or were plausible. Corrections to these were made where possible (e.g., it was clear an extra zero had been added at the end of a reported step count). Step data are typically truncated to 30,000 steps [31], which is a hypothetical cap added to step counts that are deemed implausible to achieve in a day. As other forms of activity also contributed towards step totals for the Step Count Challenge and not just walking, the Path for All plausible step count limit of 50,000 was used to exclude potentially erroneous data. Steps per day were calculated by dividing the weekly step total by the number of days reported. Step data are reported as steps per day and labelled as ’daily steps’. Many participants were not routinely collecting step counts for the week before the challenge and entered either no data (missing) or ‘zero’ steps at baseline. It was decided by the authors that the likelihood that an individual accumulates zero steps over a full week period is unlikely, and so these zero data points, such as the missing data, were excluded from the analysis. All outcome data were cleaned based on a worst-case scenario where every missing data point was coded as missing and not imputed. This was done to gain perspective on how well outcomes were reported, acknowledging that the Step Count Challenge is a real-world walking intervention. The n’s presented in text and the tables are therefore significantly lower than the number who completed the baseline survey.

Continuous baseline sample characteristics were presented as means ± standard deviation (SD) and categorical data presented as numbers and proportions. Age and gender (both categorical data) of the current sample were compared to those who took part in the Step Count Challenge using routine data collected by Paths for All. Steps and work-related outcomes are presented as means ± standard deviation, median and 25th%, 75th% percentiles. Changes in steps and work-related outcomes during the course of the Step Count Challenge were assessed using repeated measure ANOVAs for nonparametric data (Friedman’s test). Changes in steps were determined for all participants with steps data (*n* = 146, *n* = 161 and *n* = 140 at each respective time point) with a total of 447 observations. The Friedman’s test was repeated for those with no missing step data across the three time points (*n* = 73, 219 observations). The results were the same irrespective of the sample tested and so data for the entire sample only (Friedman’s test) are presented. For measures with subscales (HWQ and UWES), change data were only assessed on the overall score and not the subscales separately. Routine Path for All data on participant engagement was also provided for comparison.

## 3. Results

Of the 2042 employees who signed up to the Step Count Challenge (1206 new users and 836 previous challenge users), 246 (12%) agreed to take part in the study and completed the baseline survey. Participant demographics for the study are shown in Table 1. The average age of participants was 42.5 years old, and most were female (83%), working full time (87%) and earning under GBP 30,000 (52%). A variety of job sectors were represented with the majority from local authorities (30%) and the National Health Service (NHS, 19%). The demographics of the participants in this sample were comparable to those who registered for the challenge (75% female, 57% selecting an age bracket between 35 and 54 years of age).

Baseline step counts were 9747 ± 5050 steps per day (Table 2). For work-related outcomes, participants reported a higher-than-average baseline stress levels (16.7 ± 6.6) with 35% (*n* = 84) classed as having high stress. Mean work engagement, vigor, absorption, and dedication scores were categorized as ‘average’ compared with normative data. No relevant normative productivity data were found.

When looking at how data were reported, step data were the least well reported outcome variable at baseline with 59% (*n* = 146). Forty-two participants (17%) entered ‘zero’ steps at baseline and 24% (*n* = 58) had missing data. Two participants reported daily step counts that was more than 50,000 steps per day. These two data points occurred in week 4 and were truncated to 50,000. For work-related outcomes, the least complete data set at baseline was for work engagement (*n* = 226 of 246) and the Perceived Stress Scale the most complete (*n* = 237 of 246). There was low compliance for completion between surveys for work-related outcomes but comparable between the measures (stress 41%, productivity 42% and work engagement 44% lower number of participants between baseline and the week 4 survey). Reporting of step count data was lower than the other measures at baseline but remained similar across all three survey time points. Drop out of participants in the challenge itself was lower than that of the of the research study with only 14% not engaging in week 4 of the challenge (*n* = 281 of those who signed up) compared to 45% in the study sample (*n* = 135 on average across the four outcomes collected); shown in Table 2.

Steps and work-related outcomes during the Step Count Challenge are shown in Table 2. Mean daily steps and work-related outcomes for each time point are presented in Figure 2 and Figure 3, respectively, with the six productivity subscales for each time point shown in Figure 4. For step counts, step values increased significantly over the Step Count Challenge period (Q(2) = 41.9172, *p* < 0.001) with a mean increase in steps between baseline and week 4 of 3374 per day. For work-related outcomes, total productivity significantly increased (Q(2) = 17.9569, *p* < 0.001), perceived stress significantly decreased (Q(2) = 21.3962, *p* < 0.0001) and work engagement remained unchanged (Q(2) = 2.7420, *p* = 0.254).

**Table 2 ijerph-19-01843-t002:** Steps and work-related outcomes (stress, work engagement and productivity) during the step count challenge (*n* = 246).

	Baseline		Week 1		Week 4	
Steps	*n* = 146 (59%)	9747 ± 5050 9035 (6744, 11,714)	*n* = 161 (65%)	12,063 ± 6097 11,500 (8571, 13,783)	*n* = 140 (57%)	13,121 ± 7650 11,738 (9286, 15,669)
Total productivity ^1^	*n* = 229 (93%)	7.4 ± 1.2 7.4 (6.6, 8.2)	*n* = 147 (60%)	7.7 ± 1.1 7.9 (7.3, 8.5)	*n* = 129 (52%)	7.7 ± 1.3 8.0 (7.1, 8.7)
Stress	*n* = 237 (96%)	16.7 ± 6.6 17.0 (12.0, 21.0)	*n* = 164 (67%)	14.6 ± 7.1 14.0 (9.0, 19.0)	*n* = 139 (57%)	15.3 ± 6.5 15.0 (12.0, 20.0)
Work Engagement	*n* = 226 (92%)	3.9 ± 0.9 4.0 (3.2, 4.6)	*n* = 161 (65%)	3.9 ± 1.0 4.0 (3.3, 4.6)	*n* = 130 (53%)	3.9 ± 1.0 4.1 (3.2, 4.7)

Data presented as mean ± standard deviation, median (25th, 75th percentile); The number and percentage of those from the initial 246 who completed each measure at each time points are reported in the grey columns. ^1^ Comprises 6 variables (shown in Figure 4).

## 4. Discussion

The aim of this study was to describe (1) the process of collecting work-related outcomes as part of a real-world workplace step count challenge, and (2) report physical activity levels (step counts) and work-related outcomes (productivity, perceived stress, and work engagement) during the Step Count Challenge. The findings of this study suggest that the Step Count Challenge can have positive change on employee step count and some work-related outcomes, including stress and productivity. Importantly, this study identified several challenges when evaluating a real-world intervention.

From the perspective of conducting a research study in a real-world intervention only a small percentage of employees that enrolled in the Step Count Challenge agreed to participate in this study. Whilst the reasons for non-participation in the study were not requested, other research has suggested research burden may be a barrier with a recommendation to streamline intervention material where possible [32]. The current study was well supported and integrated into the challenge where possible by Paths for All. For example, they advertised the study through their website and sign up blog, provided incentives to participate and sent out the link to the survey on our behalf. However, it was fundamentally separate from the intervention. Future Step Count Challenge evaluation may benefit from incorporating research routinely into the intervention with permissions to use data for research purposes. This may also increase the number of participants and reduce potential research selection bias with those who are less active, less productive, more stressed, and less engaged less likely to sign up to a study addressing these factors. Even if ‘tag on’ studies like the current study were still to be conducted, increasing the amount of routine data collected by the interventionist as part of the intervention itself (such as more demographic information) would allow researchers to evaluate whether their subsample is more representative of the intervention as a whole and reduce participant burden. Gender and age were compared in the current study to the Step Count Challenge routine data which showed that the high percentage of women that participated in the study (83%) was comparable to the national data (75% female) suggesting an intervention bias over a study bias. However, caution needs to be taken as any research or routine data collection must not detract from the purpose of the intervention which is to increase physical activity levels.

There was low initial uptake and low compliance for completion between surveys for workplace outcomes reported in this study. Baseline step count data were also poorly reported where the participants were asked to report steps a week before the challenge started purely for the purpose of research. Many participants might not have been recording steps for the week previous as this was not part of the intervention. Only those who routinely use activity trackers would have been able to provide this information and those who were not might have chosen not the take part in the study. Again, incorporating research into the intervention by automatically transferring step data from the website for those who consent to participate would reduce participant burden and potentially improving the reporting of step count data. However, since the additional research baseline implemented for this current study is not currently part of the challenge, strategies such as pre-notifying participants of the need to start collecting step data earlier might be beneficial [33].

Whilst adding a period of baseline before the study led to a smaller sample of those pre-recording their steps, with regard to inferring improvements in outcomes, this baseline period was essential. At present, many real-world workplace step-counting interventions such as the Step Count Challenge do not include what would be seen in research as a ‘true’ baseline [34]. This is a period of time when a participant is recording data such as steps but is asked not to change their daily activity habits. This is again the balance between research and real-world interventions where it is not a palatable or potentially ethical message to ask people who have signed up to become more active not to start increasing their step counts in order to record a baseline measure for research purposes. However, the fact that only significant changes in outcomes were noted when compared to the baseline and not the first week of the challenge highlight the importance of incorporating a period of measurement before the intervention has begun. With many workplace step count challenges incorporating goal setting as part of their behaviour change techniques, it could be argued that participants motivation might be boosted if their step goals were calculated compared to their true baseline steps and not after they have already tried to increase their steps. If baseline periods are messaged as part of the intervention and not for research purposes per say, then this might be beneficial for both the participants, the intervention and research outcomes.

Whilst caution should be taken when interpreting change in outcomes as a result of the pre-post study design, there is indication that the Step Count Challenge results in an increase in steps and work-related outcomes; namely stress and productivity. This is supported by previous studies on the Spring Step Count Challenge that also suggest steps counts are increased [35]. Niven et al. (2021) reported a mean difference in steps per day of 906 (range 506 to 1223 steps per day) between week 1 and week 8 of the Step Count Challenge across four years of intervention delivery and 10,183 participants. This is a smaller increase than the current study where the mean difference in steps was 3374 between research baseline and week 4, although is similar to the difference between week 1 and week 4 in the current study (1058 steps). This was unexpected given the current challenge was delivered during the Autumn when seasonality would suggest a lower step count [36]. In addition, it is interesting to notice that the difference in mean daily steps between week 1 and 4 of the current study was not as great as between baseline and week 1. This may suggest that just starting such a challenge could be enough to boost step count with this level then maintained for the subsequent weeks through the additional intervention support provided.

There was also preliminary evidence that the Step Count Challenge might produce changes in work-related outcomes. Small but significant reductions in perceived stress scores were reported during the intervention. As with changes in mean daily step counts, this was most notable between the research baseline and week 1 of the intervention. This is unexpected as it is thought a lag time would exist between starting a physical activity intervention and changes in outcomes. However, the challenge involves the creation of a team prior to commencing the challenge with social support previously shown to act as buffers to occupational stress and job-related strain [37]. It may also be that people who self-selected to take part in the study were already engaged with activities to improve their health and stress which could have impacted on the results.

This study is also thought to be one of the first to report significant changes in total productivity outcomes for a workplace walking intervention. Puig-Ribera et al. 2008 conducted a randomized control trial of 70 employees to assess the impact of two walking interventions on work productivity using the Work Limitations Questionnaire [38] over a nine week period [21]. The walking interventions included a “walking routes” group who were provided with a map of examples walks in their locality and were asked to complete a minimum of 15 min of continuous, brisk walking every work day. The “walking while working” group were encouraged to accumulate step counts by replacing sedentary work tasks with standing and walking, such as walk and talk meetings. Findings suggested no changes in step count or work performance for either intervention compared to the control group. However, when the intervention data were pooled including both intervention groups, significant increase in steps were reported for those classed as low active at baseline and a positive but not significant change in productivity profiles. Given the importance of work-related outcomes on convincing employers to provide the time for employees to be active and the easy of walking as an intervention strategy, this suggested the importance of collecting work-related outcome data from workplace walking interventions.

A strength of this study is that it evaluated a sustainable, real-world workplace step count challenge and investigated changes in outcomes of importance to workplaces. In addition, this study has provided details on the process of conducting physical activity research in a real-world intervention which has significant implications for future delivery. With the desire to ‘evaluate’ rather than ‘research’ the intervention came a significant limitation of the study with regard to reporting changes in outcomes such as the pre-post design and lack of control group. Future research in this area could look at using more robust study designs. However, the logistics of this need to be considered as not to interfere with the delivery of the Step Count Challenge or similar programs as the primary objective of such real-world interventions is to deliver an intervention and change physical activity levels at a population level. Whilst it is unlikely researchers will be able to conduct a true randomized control trial on such interventions, investigating ways to deliver a more robust study design and control group should be investigates. For example, this could be achieved is by running bespoke challenges outwith the main intervention delivery and using a stepped wedge cluster randomized control design so all workplace still receive an intervention.

Another limitation is the high step count of the participants at baseline (which were nearing 10,000 steps per day) and the high percentage of female participants. Given that the baseline step counts and gender split for this study is similar to the report for the national data [35], it is likely that these are considerations not only for research but for the Step Count Challenge itself. Using strategies to engage more men and less active employees in such interventions should be a key target of future challenges. It is also imperative that researchers and real-world physical activity delivery agents work together to try and incorporate research and routine data collection more seamlessly into real-world interventions. Suggestions are made within this paper as to how to bridge the gap between the two agendas of a physical activity delivery agency and researchers with a detailed table of lessons learnt and future recommendations provide below.

## 5. Conclusions

The findings from this study suggest that whilst there are numerous challenges to conducting research in a real-world physical activity intervention such as the Step Count Challenge, there are strategies that can be used to overcome these. These include streamlining and integrating research and routine data collection into real-world interventions, including providing true baseline. Preliminary evidence from this study also suggests positive changes in step counts and work-related outcomes such as stress and productivity. This warrants further investigation and applying the recommendations from this paper on conducting more robust research without influencing the intervention.

Lessons learnt and future recommendations for researchers and interventionists:Link research into real-world interventions more explicitly and seamlessly
Interventions to streamline research studies into interventions more seamlessly where possible as not to be seen as an add onInterventionists to incorporate more data collected routinely such as demographics, physical activity data with permissions for research purposesInterventionists to work researchers to automictically transfer routinely collected data (if online) for research purposesResearchers to provide interventionists with a toolbox of short, validated measures of workplace outcomes for routine use in real-world interventions
Collect a true baseline
Interventionists to incorporate ‘true baseline’ period as part of the interventionResearchers to pre-warm participants about the need to record true baseline if not part of the intervention
Assess appropriateness of a robust study design
Researchers and interventionists to work closely to implement the most robust but appropriate research design
Develop positive action recruitment strategies
Interventionist to look at ways to attract a more equal gender balance and those who are inactiveResearchers to assess sampling strategies to address studies with inequalities

## Figures and Tables

**Figure 1 ijerph-19-01843-f001:**
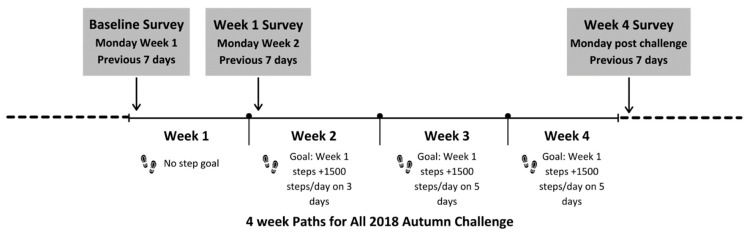
Description of data collection from baseline to week 4 survey mapped onto the Step Count Challenge weeks.

**Figure 2 ijerph-19-01843-f002:**
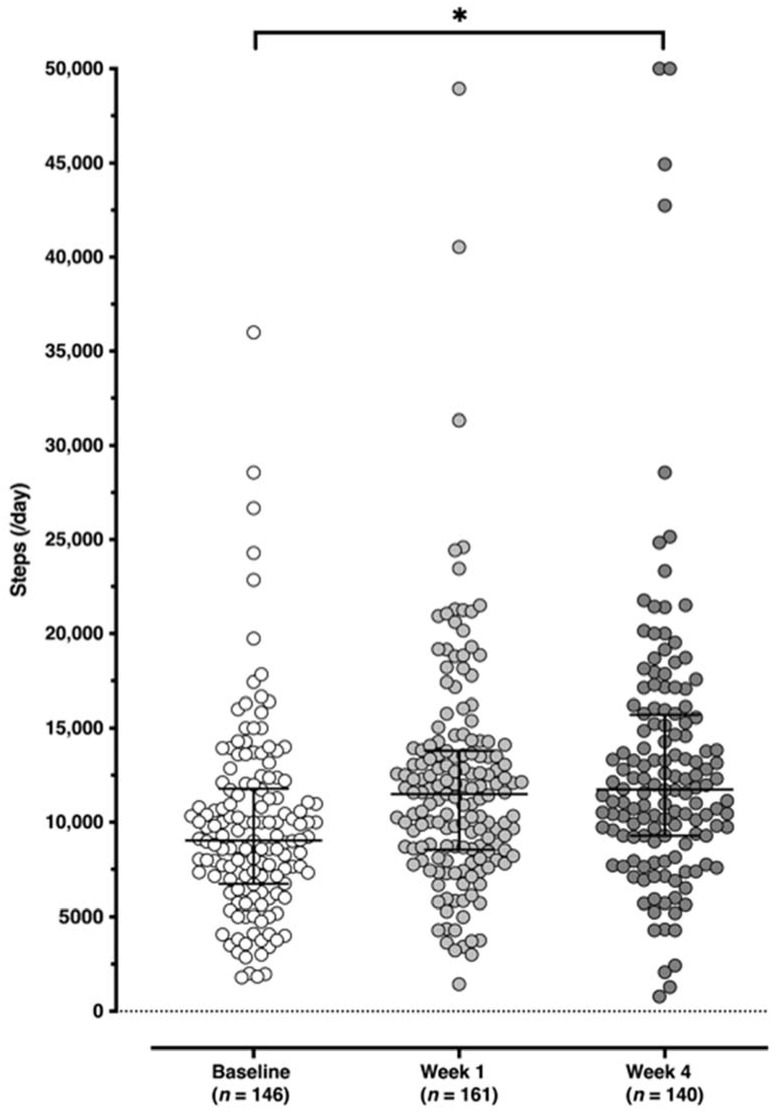
Mean daily steps for each time point (baseline, week 1 and week 4). * *p* < 0.001.

**Figure 3 ijerph-19-01843-f003:**
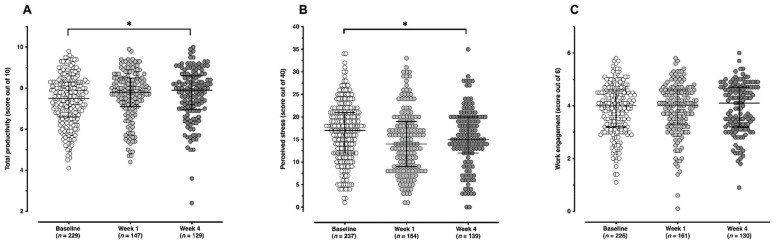
Work-related outcomes (**A**) total productivity, (**B**) perceived stress, (**C**) work engagement at each time point (baseline, week 1 and week 4). * *p* < 0.001. Total productivity is further broken down into its six subscales in Figure 4.

**Figure 4 ijerph-19-01843-f004:**
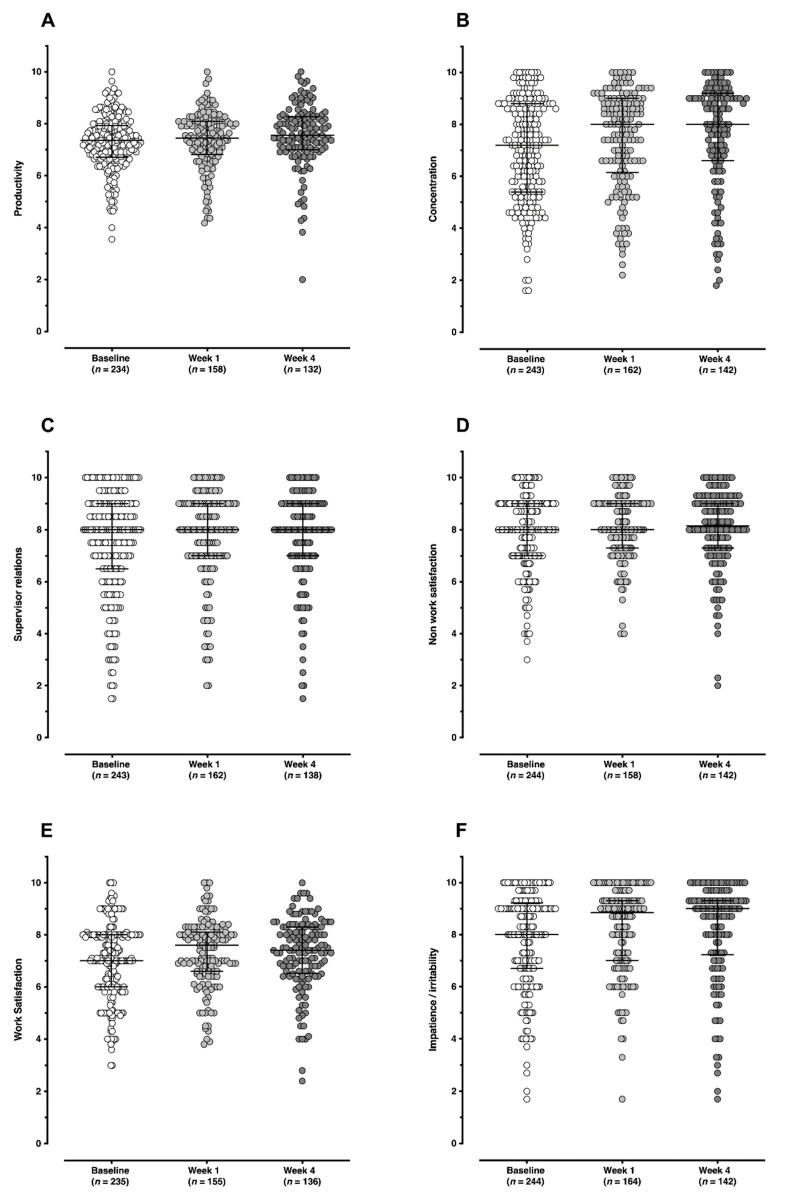
Six subscales that form the total productivity score at each time point (baseline, week 1 and week 4). Productivity (**A**), concentration (**B**), supervisor relations (**C**), non-work satisfaction (**D**), work satisfaction (**E**), impatience/irritability (**F**). Scores closer to 10 indicate desirable work productivity outcomes. Changes in subscales were not assessed with formal statistical testing.

**Table 1 ijerph-19-01843-t001:** Participant characteristics at baseline.

Variable	Baseline Measurement (*n* = 246)
**Age** (Mean ± SD, years)	42.5 ± 11.1
**Gender**	
Male	41 (17)
Female	204 (83)
**Employment Status** (n (%))	
Full-time	213 (87)
Part-time	32 (13)
**Annual Income** (n (%))	
Less Than GBP 20,000	41 (17)
GBP 20,001 to 30,000	87 (35)
GBP 30,001 to 40,000	63 (26)
Over GBP 40,001	54 (22)
**Work Sector** (n (%))	
NHS	46 (19)
Private	29 (12)
Local Authority	74 (30)
Higher education	24 (10)
Third Sector	31 (12)
Other public sector	42 (17)
**Expected work hours** (mean ± SD)	37.1 ± 0.9
**Average hours across the challenge** (mean ± SD)	35.5 ± 7.2

## Data Availability

The data presented in this study are available on request from the corresponding author.

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
