# Peer review of "Measuring Productivity, Perceived Stress and Work Engagement of a Nationally Delivered Workplace Step Count Challenge"

_ijerph, 2022, doi:10.3390/ijerph19031843_

Round 1

Reviewer 1 Report

This reports on an interesting piece of research that reveals some evidence of an increase in self-reported productivity at work with increased steps, as part of a national step count challenge in Scotland, alongside a reduction in stress and almost no impact on work engagement.  However the paper as submitted is not written with sufficient clarity to be accepted in its current form. 

The inadequacy of the paper starts with the title that does not quite match the research.  The paper does not cover "Evaluation of a nationally delivered workplace step count challenge: Productivity, perceived stress and work engagement".  What in fact it does is use the opportunity of a national Step Count Challenge in Scotland, that signed up 2042 employees at the time of the research, to test the association between increased stepcount and measures of productivity, perceived stress and work engagement, for a small sample of participants who agreed to take part in the study (N=246), most of whom are females working in the public sector.  The paper does make some useful recommendations for extending baseline measurements in conducting the national Step Count Challenge in future.

The paper needs careful editing throughout to ensure that it communicates exactly what was done, why, and the implications for the work of researchers and other professionals who are committed to finding ways to encourage greater levels of physical activity.  It also needs careful attention to correction of the numerous grammatical errors, inclusion of omitted words, and elimination of unnecessary words. There may also be some confusion about the sample size.  While the total number of participants is noted as 246, complete results are reported for considerably fewer, that may be N=129 (Table 2).  The text does not clarify if this is the case or whether the N's refer to completion of each question, making the total N for respondents who completed all 12 data points somewhat fewer.

p3, line 136 refers to data being based on 7-day recall, that would appear to be highly unreliable.  Clarification is needed.

Clear and careful communication of all results and discussion must be improved in any future submission.  The last paragraph for example contains the following sentence: "However, whilst hard to conduct this sort of research with the same degree of rigor that is expected from traditional academia, there is in no doubt that this of research is needed".  Apart from the fact that words are missing and 'rigor' is misspelt for UK English, the assertion that this research is needed should be argued rather than just stated within a scientific paper.

I would recommend that the paper be rewritten and more carefully supervised and edited by all co-authors before resubmission. 

Author Response

On behalf of the authors I would like to thank the reviewers for their time reviewing our paper. Please find below our response and record of changes make to the original submission.

Reviewer 1

This reports on an interesting piece of research that reveals some evidence of an increase in self-reported productivity at work with increased steps, as part of a national step count challenge in Scotland, alongside a reduction in stress and almost no impact on work engagement.  However the paper as submitted is not written with sufficient clarity to be accepted in its current form. 

The inadequacy of the paper starts with the title that does not quite match the research.  The paper does not cover "Evaluation of a nationally delivered workplace step count challenge: Productivity, perceived stress and work engagement".  What in fact it does is use the opportunity of a national Step Count Challenge in Scotland, that signed up 2042 employees at the time of the research, to test the association between increased stepcount and measures of productivity, perceived stress and work engagement, for a small sample of participants who agreed to take part in the study (N=246), most of whom are females working in the public sector.  The paper does make some useful recommendations for extending baseline measurements in conducting the national Step Count Challenge in future.

The title has been changed to Measuring productivity, perceived stress and work engagement of a nationally delivered workplace step count challenge.

The paper needs careful editing throughout to ensure that it communicates exactly what was done, why, and the implications for the work of researchers and other professionals who are committed to finding ways to encourage greater levels of physical activity.  It also needs careful attention to correction of the numerous grammatical errors, inclusion of omitted words, and elimination of unnecessary words.

The paper has been checked for errors and edited accordingly.

There may also be some confusion about the sample size.  While the total number of participants is noted as 246, complete results are reported for considerably fewer, that may be N=129 (Table 2).  The text does not clarify if this is the case or whether the N's refer to completion of each question, making the total N for respondents who completed all 12 data points somewhat fewer.

In the text we state the n for each variable and what they represents. For example in line 222 we state:

Of the 2042 employees who signed up to the SCC (1206 new users and 836 previous challenge users), 246 (12%) agreed to take part in the study and completed survey one.

This is clear that the 246 is the number who complete survey one. Table 2 also says in the footnote:

The number and percentage of those from the initial 246 who completed each measure at each time points are reported in the grey columns.

However for clarity, this has been added to Line 205 in the methods: The n’s presented in text and the tables are therefore lower than the number who completed the baseline survey.

p3, line 136 refers to data being based on 7-day recall, that would appear to be highly unreliable.  Clarification is needed.

The 7 day recall is what is used in the validated measures. It is not possible to change the validated questionnaires recall period. This sentence has been changed reflect the need to keep the validated questionnaire:

Line 137: Each time point aimed to capture data for the previous 7 days in line with the time periods specified in the majority of validated questionnaires used.

Clear and careful communication of all results and discussion must be improved in any future submission.  The last paragraph for example contains the following sentence: "However, whilst hard to conduct this sort of research with the same degree of rigor that is expected from traditional academia, there is in no doubt that this of research is needed".  Apart from the fact that words are missing and 'rigor' is misspelt for UK English, the assertion that this research is needed should be argued rather than just stated within a scientific paper. I would recommend that the paper be rewritten and more carefully supervised and edited by all co-authors before resubmission. 

The results and discussion have been edited to improve communication. There is also a table with a summary of the results for the reader to view without reading the whole paper. The sentence referred to has been deleted. The latest version has be double checked and edited by co-authors.

Reviewer 2 Report

This is a well-written study with important implications, which examines whether the Step Count Challenge (SCC) in the workplace increases the numbers of steps walked by employee along with their productivity, stress, and work engagement. Although the lack of a control group and the bias of participants are extremely significant problems in this study, they are adequately addressed in the discussion section.

I would like to extend appreciation to the authors for their contribution to the field of increasing physical activity in a real-world intervention. However, I have added a few comments below for the authors to review.

  1. Line 62: The first author's name + et al. should be the appropriate format of citing the reference.
  2. Lines 103-112: I believe that it would be helpful to the reader if the number of recruits for the participants is listed in this section.
  3. Line 133: The readability would be enhanced if the survey section is divided into smaller sections and each variable has its own item title.
  4. Line 210-212: Since this is the repeated measurement of an individual, do you not agree that a multi-level analysis would be more appropriate? If possible, consider conducting a multi-level analysis. I would also recommend that the authors perform a statistical analysis of the subscale in Fig. 4. In addition, I believe that examining the increasing trend (conducting trend test) as well as the difference among the groups would be effective ways to show the impact of the SCC.
  5. Tables 1 and 2: I believe the readability would be enhanced if the letters were left-justified, and the numbers were right-justified (align decimals, etc).

Author Response

On behalf of the authors I would like to thank the reviewers for their time reviewing our paper. Please find below our response and record of changes make to the original submission.

Reviewer 2

This is a well-written study with important implications, which examines whether the Step Count Challenge (SCC) in the workplace increases the numbers of steps walked by employee along with their productivity, stress, and work engagement. Although the lack of a control group and the bias of participants are extremely significant problems in this study, they are adequately addressed in the discussion section.

I would like to extend appreciation to the authors for their contribution to the field of increasing physical activity in a real-world intervention. However, I have added a few comments below for the authors to review.

  1. Line 62: The first author's name + et al. should be the appropriate format of citing the reference.

This has been changed to et al.

  1. Lines 103-112: I believe that it would be helpful to the reader if the number of recruits for the participants is listed in this section.

This has been added. Line 112 now reads: For the present study, 2042 employees signed up to the SCC (1206 new users and 836 previous challenge users).

  1. Line 133: The readability would be enhanced if the survey section is divided into smaller sections and each variable has its own item title.

This section has been rewritten to ensure all outcomes have their own subsubsection.

  1. Line 210-212: Since this is the repeated measurement of an individual, do you not agree that a multi-level analysis would be more appropriate? If possible, consider conducting a multi-level analysis. I would also recommend that the authors perform a statistical analysis of the subscale in Fig. 4. In addition, I believe that examining the increasing trend (conducting trend test) as well as the difference among the groups would be effective ways to show the impact of the SCC.

We appreciate the reviewers thoughts on this and we discussed this at length within the research team. We conducted several different analysis and found very similar results regardless. We opted for this more simplistic approach as it reflected the research design better as quite often researchers try and use analysis as a way of compensating for less rigorous research designs. Since the main aim was to report on the process as opposed to just the change in outcomes, we believe the current analysis is the best suit for the data. In addition, for a similar reason we did not conduct analysis on the subscales in Fig 4 and just reported visuals of the trends which we do for each variable to show the trends in the data.

  1. Tables 1 and 2: I believe the readability would be enhanced if the letters were left-justified, and the numbers were right-justified (align decimals, etc).

This change has been made.